# Current Sensor with Optimized Linearity for Lightning Impulse Current Measurement

**DOI:** 10.3390/s25144516

**Published:** 2025-07-21

**Authors:** Wenting Li, Yinglong Diao, Feng Zhou, Zhaozhi Long, Shijun Xie, Jiawei Fan, Kangmin Hu, Zhehao Wang

**Affiliations:** 1China Electric Power Research Institute, Beijing 100192, China; diaoyinglong@126.com (Y.D.); zy397037172@163.com (F.Z.); longzhaozhi@epri.sgcc.com.cn (Z.L.); fanjw@epri.sgcc.com.cn (J.F.); wangzhehao@epri.sgcc.com.cn (Z.W.); 2State Grid Sichuan Electric Power Research Institute, Chengdu 610041, China; sj-xie@163.com

**Keywords:** lightning impulse current, scale factor, linearity

## Abstract

Impulse current measurement technology is widely used in various applications, including lightning protection monitoring in power systems, welding current measurement in aircraft and shipbuilding industries, as well as high-current measurement in pulsed power systems. With the advancement of industrial technology, the measurement range of impulse currents has continuously expanded, reaching levels as high as mega-amperes (MA). The calibration of the scale factor for impulse current measurement devices is determined through comparison with standard measurement devices. Developing high-accuracy impulse current measurement devices and accurately judging their characteristics are prerequisites for ensuring the precise calibration of impulse current values. This paper introduces two different types of high-impulse current measurement devices. Experimental studies were conducted on the scale factor and response characteristics of the sensors. The scale factor extension calibration method for sensors under high currents of more than 100 kA has also been introduced. Test results indicate that the developed impulse current measurement devices can serve as standard measurement devices for high impulse current measurement.

## 1. Introduction

Impulse current measurement technology is widely applied in power systems, meteorological research, industrial manufacturing, and power measurement of pulse power sources. The measurement accuracy of impulse current directly affects the safe and stable operation of power systems, relates to the precise control of industrial production processes, and determines the accurate power calculation of high-power pulse current sources [1,2,3,4,5]. Shunt, Rogowski coils, and magneto-sensitive sensors are commonly used for lightning impulse current measurements. Shunts, designed with a coaxial structure, offer excellent response characteristics and shielding performance, were usually adopted as standard measurement devices. Rogowski coils and magnetic field sensors, operating on non-contact measurement principles, inherently offer wider current measurement ranges and have gained extensive application in field measurement scenarios [6,7,8,9].

With the advancement of various industries, the demand for measuring the amplitude of impulse currents continues to rise. In the pulse power field, impulse current amplitudes can reach several megaamperes (MA), placing higher demands on the response characteristics, stability, and thermal tolerance performance of impulse current measurement devices [10,11]. The calibration of impulse current measuring devices by comparison with national/international standard lightning impulse current measurement systems constitutes a critical approach to ensuring measurement accuracy. However, internationally recognized standard systems have a rated current not exceeding 100 kA [12,13], which cannot meet full-scale calibration requirements. Supplementary linearity calibration tests were needed for the accurate measurement of impulse current value. Consequently, the development of high-accuracy, large-current linearity measurement devices to evaluate the nonlinearity characteristics of high-current impulse measuring devices under large currents has become a critical aspect of high-current impulse measurement technology [14].

Rogowski coils and magnetic field sensors are always used as high impulse current measurement sensors, owing to their advantages of non-contact connection, negligible heating effects, and independence from rated current limitations. Rogowski coils encompass both magnetic-core structures and air-core measurement coils. When measuring high currents exceeding 100 kA, air-core Rogowski coils are generally employed to prevent distortion of measurement results caused by magnetic core saturation [15,16,17,18,19].

Magnetic field sensor based on magneto resistive chips—including TMR (Tunnel Magnetoresistance), GMR (Giant Magnetoresistance), and Hall effect types—are now widely adopted in current measurement technologies. However, such sensors remain unsuitable as standard current measurement devices due to their significant susceptibility to ambient temperature and humidity variations [20,21,22,23,24]. Similarly, fiber-optic magnetic field sensors face stability limitations [25,26,27]. Both of the above magnetic field sensors are currently primarily applied to AC/DC current measurements. When utilized for impulse current measurements, further expansion of their frequency bandwidth characteristics is required. Consequently, this study primarily focuses on magnetic field sensors based on inductive coil structures.

During high-current impulse discharges, intense electromagnetic interference is generated in the surrounding space. This necessitates that current sensors possess exceptional shielding capabilities and interference immunity. In-depth investigation into the sensor’s shielding structure and circuit parameter design is required [28,29] to ensure accurate current measurements across the full-scale range. For the developed current sensors, determining their linearity parameters accurately constitutes another key research focus.

This study develops high-precision air-core Rogowski coils and magnetic field sensors based on Printed Circuit Board coil configurations to achieve accurate measurement of impulse currents at the hundred-kA level. First, the structures and operational principles of both sensors are analyzed, with corresponding structural designs and electrical circuit parameters proposed. Subsequently, key parameters including frequency bandwidth and sensitivity are analyzed and calculated. An impulse high-current characteristic test platform is established to validate the accuracy of the determined linearity parameters, and an experimental methodology for large-current measurements using these sensors is presented.

## 2. Lightning Impulse Current Measurement Sensor

### 2.1. Rogowski Coil

The non-intrusive installation of the Rogowski coil determines that it is not directly affected by the thermal effects of high current. The factors affecting its measurement accuracy mainly include bandwidth limitations, eccentricity effects, winding uniformity, leakage flux, and external electromagnetic interference. Under different test currents, the electromagnetic interference caused by transient discharge varies. Therefore, a Rogowski coil with strong shielding capability should be selected for linearity testing.

The structure diagram of the Rogowski coil is shown in Figure 1. Taking the cross-sectional area micro-element dS, the magnetic flux passing through the micro-element of this area is shown in Formula (1).

The magnetic flux through a single-turn current coil is(1)$$d\phi =B\cdot dS=\frac{\mu_{r}\mu_{0}i}{2\pi y}dxdy,$$
where *i* is the current to be measured, *μ*_0_ is the vacuum permeability (4π × 10^−7^ H/m), *μ*_r_ is the relative magnetic permeability of magnetic core material, and *μ*_r_ = 1 when the winding skeleton is made of insulating material, *y* is the coordinate value in the Y-direction on the core interface.

The magnetic flux ϕ linked by a single turn coil is(2)$$\phi =\iint_{S}\frac{\mu_{0}\mu_{r}i}{2\pi y}dxdy=\frac{\mu_{0}\mu_{r}i}{2\pi }\int_{-\frac{h}{2}}^{\frac{h}{2}}\int_{a}^{b}\frac{1}{y}dxdy=\frac{\mu_{0}\mu_{r}hi}{2\pi }\ln\frac{b}{a}$$

The induced electromotive force of the current coil is(3)$$e(t)=-N\frac{d\phi }{dt}=-\frac{N\mu_{0}\mu_{r}h}{2\pi }\ln\frac{b}{a}\cdot \frac{di}{dt}dxdy,$$
where *N* is the number of turns of the coil.

In order to enhance the sensitivity of the sensor and ensure its uniform winding, a current coil based on a multi-layer Printed Circuit Board (here inafter referred to as PCB) structure design was chosen [10,11], as shown in Figure 2. The top and bottom layers of the multi-layer PCB form a Faraday cage shield to block interference from external electric fields. The two copper-clad shielding layers are interconnected through ground wire vias, allowing charge flow for the neutralization of reverse electric fields from external sources. The internal structure and appearance of the Rogowski coil are as depicted in the figure. The coil is equipped with an analog integration circuit, with a designed scale factor of 1.25 V/kA.

The characteristic parameters of the designed Rogowski coil were determined through theoretical analysis and simulation calculations, as shown in Table 1. The simulation results demonstrate that the designed Rogowski coil meets the requirements for measuring lightning current waveforms of (4/10) μs and (8/20) μs.

### 2.2. Magnetic Field Sensor

The magnetic field sensor developed by CEPRI is based on a constant-speed Archimedean spiral coil. Similarly to the Rogowski coil, it measures current based on the magnetic field, and the difference is that the sensing turns of an Archimedean coil are arranged in the same plane, and the conductor under test does not pass through the center of the Archimedean coil. The spiral coil is also printed on a PCB circuit board to improve the distribution uniformity of the coil. The voltage induced by the magnetic sensor is proportional to the change in the current being measured. The total induced electromotive force of the Archimedean spiral coil is the result of the superposition of the induced electromotive forces from multiple arc segments and the coil’s center. As a result, its coil area utilization efficiency is higher.

The positional relationship between the magnetic induction coil and the current-carrying conductor is shown in Figure 3. To maximize the magnetic flux passing through the induction coil, the plane of the coil must be perpendicular to the direction of the magnetic field generated by the current.

The time-domain expression of the impulse current *i*(t) is set as(4)i(t)=Ia(t),
where *I* is the peak value of the impulse current, and *a*(t) is the functional expression of the unit impulse current.

For ease of derivation, assume that the plane of the Archimedean spiral lies in the XOY plane, with its starting point coinciding with point O. The Cartesian coordinate function of the Archimedean spiral is(5)$$\left\{\begin{array}{l} x=r\cos\theta ,\\ y=r\sin\theta ,\\ r=b\theta , \end{array}\right.$$*R* = *a* + b*θ*.(6)

As shown in Figure 4, *r* is the distance from a point on the spiral to the fixed point. When *θ* = 0, a is the distance from the starting point of the spiral to the origin of the polar coordinates. *θ* is the total rotation angle (in radians) of the spiral.

To further improve the sensitivity of the Archimedean spiral coil, a double-layer series configuration is employed, effectively doubling its sensitivity. Figure 5 illustrates the schematic diagram, and the physical prototype of the magnetic field sensor based on the Archimedean spiral coil.

The direction of the magnetic flux density is perpendicular to the plane of the spiral. Therefore, the expression for the magnetic flux density at any point P_1_(rcosθ, rsinθ) on the double-layer spiral is(7)$$B(t)=\frac{\mu_{0}I\cdot a(t)}{2\pi (d-r\cos\theta )}=\frac{\mu_{0}I\cdot a(t)}{2\pi (d-r\cos\theta )}.$$

Then, the induced electromotive force dE on the infinitesimal arc at point P_1_ (in Figure 4) is(8)$$dE=\frac{dB^{\prime }(t)}{dt}dS.$$

Thus, the induced electromotive force E for an Archimedean spiral coil of double layer can be calculated as(9)$$\begin{array}{ll} E & =2\times \frac{d}{dt}\iint_{S}B(t)dS=2\times \frac{d}{dt}\int_{0}^{2n\pi }\int_{0}^{b\theta }B(t)rdrd\theta \\ & =2\times \frac{\mu_{0}I}{2\pi }\cdot \frac{da(t)}{dt}\int_{0}^{2n\pi }\int_{0}^{b\theta }\frac{r}{\left(d-r\cos\theta \right)}drd\theta \\ & =\frac{\mu_{0}I}{\pi }\cdot \frac{da(t)}{dt}\int_{0}^{2n\pi }\int_{0}^{b\theta }\frac{r}{\left(d-r\cos\theta \right)}drd\theta \end{array}.$$

Simulation calculations of the Archimedean spiral coil were performed based on the designed parameters of the Archimedean coil, and the frequency response characteristics of the Archimedean coil were obtained through analysis, as shown in Figure 6. The frequency range spans from a minimum of 3 Hz to a maximum of 21 MHz. The frequency band characteristics indicate that its high-frequency performance fully meets the requirements for measuring impulse lightning currents and its low-frequency performance is slightly inferior compared to the Rogowski coil.

For a magnetic field sensor, when its measurement range of magnetic field values is fixed, the linearity of its scale factor (the variation in the scale factor under different magnetic field values) is the same. Therefore, when the magnetic field sensor is placed close to the current conductor under test, the linearity of the scale factor over a wide magnetic field range can be determined by comparing it with a standard shunt. For example, test 1: when the magnetic field sensor is placed 10 mm away from the current conductor, a comparison with a 100 kA standard shunt can determine the corresponding scale factor and its variation for each magnetic field value within a test range of up to 89.5 A/m. Test 2: When the current under test increases to 500 kA, the distance between the magnetic field sensor and the current conductor can be increased to 30 mm or more. Under a test current of 100 kA, the measured magnetic field value and the corresponding scale factor of the sensor can be determined. Based on the linearity of the scale factor established in test 1, the scale factor corresponding to each current point from 100 kA to 500 kA can be determined for the magnetic field sensor. Based on the scale factor under 500 kA, the magnetic field sensor can serve as a high-current measurement standard for currents exceeding hundreds of kA for other current measurement devices. Compared with the Rogowski coil, the Archimedean magnetic field sensor theoretically can provide better measurement accuracy since it does not require curve fitting or extrapolation to determine its scale factor.

As a single magnetic field sensor may be susceptible to spatially interfering electromagnetic fields, by designing a ring-shaped insulating support framework and arranging multiple sensor arrays, the effects of spatial co-directional electromagnetic interference can be effectively mitigated, as shown in Figure 7. For the array structure of the Archimedean magnetic field sensor construction, a circular (annular) sensor mounting framework must be machined. This framework should feature uniformly distributed arrayed mounting holes. Each mounting hole should incorporate multiple fixation slots. These slots allow flexible adjustment of the sensor’s radial distance from the central conductor under test. The output voltage signals from all magnetic field sensors must be summed. The aggregated signal is connected to the output terminal via coaxial cable. The output is then fed into a data acquisition system for multi-array magnetic field measurements.

## 3. 100 kA Lightning Impulse Current Test

### 3.1. Test of Rigid Rogowski Coil

(1) Scale factor calibration

Figure 8 shows the Schematic Diagram for the scale factor calibration for the Rogowski coil. The impulse current generator employs an energy storage capacitor with C = 6 μF, an inductor with L = 3.6 μH, and a resistor R = 0.73 Ω. Simulation results demonstrate that this configuration can generate an impulse current waveform with time parameters *T*_1_(Front time)/*T*_2_(time to half value) of 3.97/11.03 μs. The experimental layout diagram is shown in Figure 9.

To ascertain the scale factor and linearity parameters of the developed Rogowski coil, two standard shunts were selected for measuring (4/10) μs and (8/20) μs lightning current waveforms, with rated currents of 40 kA and 100 kA, respectively. The peak current measurement uncertainty is 0.4% (*k* = 2), while the time parameter measurement uncertainty is 1.0% (*k* = 2). The calibration test current value is from 5 kA to 40 kA with the 40 kA standard shunt. The calibrated scale factor of the Rogowski coil is reflected by the variation in scale factor under different test current values, as shown in Figure 10. The calibration waveform is as shown in Figure 11. Based on the determined scale factor, the fitting curve formula for calibrating the scale factor y is established asY = 9 × 10^−5^x + 1.2493.(10)

The 100 kA standard shunt was then used to calibrate the scale factor of the Rogowski coil within the 40 kA–100 kA range. The calibration results are shown in Table 2. During the calibration tests, the Rogowski coil’s scale factor was preset to 1.25 V/kA, and each set of experimental data was based on 10 repeated measurements. By comparing the actual calibrated scale factors with the fitted calibration scale factors determined using Equation (4), it was found that the deviation of the calibrated scale factors at each current point within the 40 kA–100 kA range did not exceed 0.1%. This confirms the excellent linearity of the developed Rogowski coil, and the fitted scale factor can be reliably applied to determine the scale factor beyond the 100 kA current range.

The relative error of two measurement devices is as follows: the relative error of front time is within 1%, and that of the half-peak time is within 0.5%, indicating that the developed Rogowski coil exhibits excellent response characteristics.

(2) Step response experiment

Based on the test circuit constructed in Figure 12, a step response test was conducted on the Rogowski coil to evaluate its frequency response characteristics. The step waveform generator can produce a square wave current with a pulse width of 1 μs and an amplitude of 100 A based on the discharge of the voltage on the coaxial cable. Figure 13 shows the measured step response waveform of the Rogowski coil, where the green curve represents the direct output signal from the Rogowski coil, and the blue waveform represents the output signal after integrating the Rogowski coil’s output signal.

The step response data of the Rogowski coil yielded a calculated partial response time of 83.45 ns, with an experimentally measured response time of 78.43 ns and an overshoot of 1.74%. The settling time was 380 ns. According to standard requirements, the coil is capable of measuring impulse current waveforms with rise times shorter than 1 μs.

(3) Short term stability

A 100 kA impulse current waveform was applied to the Rogowski coil test circuit 20 times consecutively with 3 min intervals between measurements. After the test, the scale factor of the Rogowski coil was measured again. The variation between the two scale factors represents the short-term stability of the Rogowski coil. The scale factor of the Rogowski coil changed by 0.11% before and after the test.

(4) Measurement Uncertainty Evaluation

Based on the above test results, the measurement uncertainty of the Rogowski coil was evaluated. Table 3 lists the uncertainty components of the calibration factor measurement results for the Rogowski coil. Within the 100 kA current range, the relative measurement uncertainty of the calibration factor for the Rogowski coil is *U*_rel_ = 0.52% (*k* = 2).

### 3.2. Test of Magnetic Field Sensor

Figure 14 illustrates the schematic diagram for the scale factor calibration test of the magnetic field sensor with a standard shunt. The magnetic field sensor is mounted on a fixture 10 mm away from the current-carrying rod, and the measured signals are transmitted to the digital recorder through coaxial cables. The test is performed at five current at 20 kA, 40 kA, 60 kA, 80 kA, and 100 kA. The error between the shunt and the Rogowski coil is evaluated by the average test results by repeating the test five times. The experimental layout diagram of the magnetic field sensor is as shown in Figure 15.

Fix the magnetic field sensor at a distance of 10 mm from the current-carrying conductor. Apply a current of 100 kA to the conductor, the corresponding magnetic field measurement range for the magnetic field sensor is 89.5 A/mm according to the simulation based on Electromagnetic simulation software Ansoft 16.0.3. The picture of the simulation results is as shown in Figure 16.

Figure 17 displays the calibrated scale factor of the magnetic field sensor under different currents. The scale factor of the magnetic field sensor rises as the test current increases. The scale factor changes by about 0.3% from 20 kA to 100 kA, with similar patterns for both positive and negative polarities.

When the actual current to be measured increases further, the distance between the magnetic field sensor and the current-carrying conductor can be increased to ensure that the measured magnetic field remains within the calibrated scale factor range of the sensor. This enables the measurement of scale factors for currents exceeding 100 kA.

## 4. Conclusions

To achieve an accurate measurement of impulse currents at the hundred-kA level, this paper presents two distinct structural configurations of impulse high-current measuring devices.

The developed Rogowski coil employs a multi-layer PCB circuit design, enhancing measurement sensitivity while maintaining electromagnetic interference immunity. Optimizing the matching of circuit parameters effectively expands its measurement bandwidth. Experimental results demonstrate a step response rise time of 83 ns and settling time of 380 ns, fully satisfying rapid impulse current measurement requirements. Within the 100 kA range, calibration against a standard shunt determined its scaling factor with linearity better than 0.3% and peak current relative measurement uncertainty of 0.52%. For currents above 100 kA, curve fitting establishes linearity parameters of the scaling factor, with accuracy experimentally verified across the 50 kA to 100 kA range.

The developed magnetic field sensor employs a double-layer Archimedean coil design, offering significantly reduced size and more convenient installation compared to Rogowski coils. Depending on field test conditions, measurements can be performed using either a single sensor or multiple sensors arranged in an array configuration. Calibration against a standard shunt determines the scale factor within a specified magnetic field range and establishes its linearity. When tested at currents between 20 kA and 100 kA, the sensor demonstrates linearity better than 0.2% within the corresponding magnetic field range. For currents exceeding this range, increasing the distance between the magnetic field sensor and the current-carrying conductor under test allows recalibration of the scaling factor within the rated current limits of the standard shunt. This approach, combined with the predetermined linearity parameters, enables impulse current measurements at substantially higher current levels.

Any of the aforementioned scale factor calibration methods can be selected to satisfy the accuracy requirements. In calibration tests, appropriate calibration methods and standard equipment can be selected based on actual needs and existing test conditions.

## Figures and Tables

**Figure 1 sensors-25-04516-f001:**
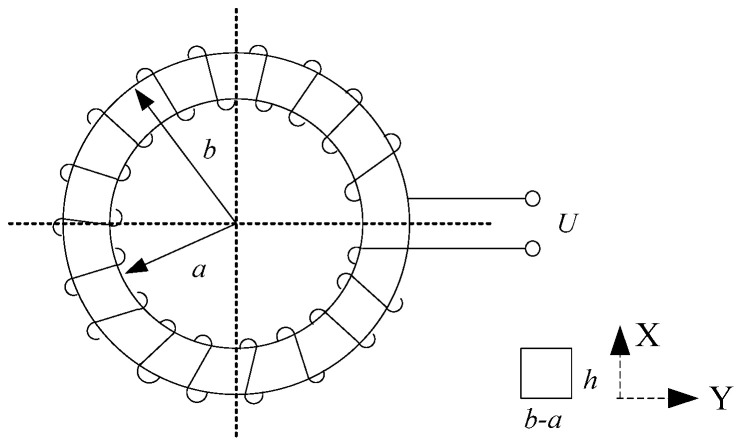
Diagram of the Structure of a Roche Coil. a: inner radius of the coil, b: outer radius of the coil, h: height of the coil.

**Figure 2 sensors-25-04516-f002:**
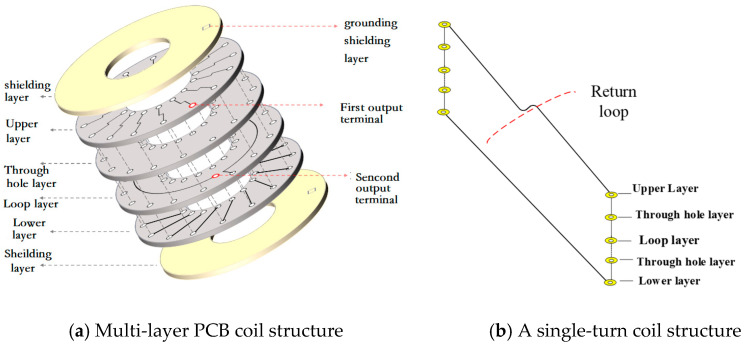
Schematic diagram of multi-layer PCB Rogowski coil.

**Figure 3 sensors-25-04516-f003:**
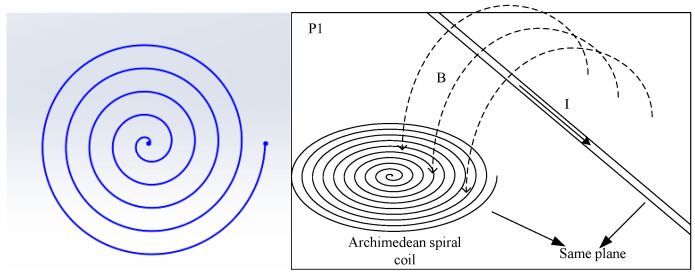
Relationship between coil and current coupling.

**Figure 4 sensors-25-04516-f004:**
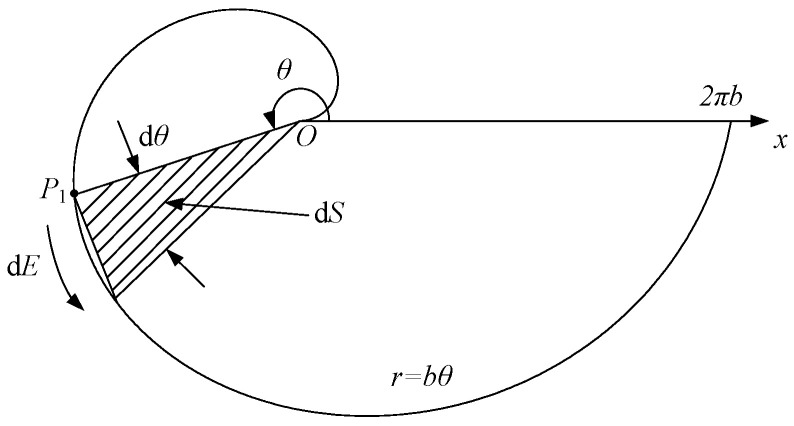
Receiving area of coil magnetic field.

**Figure 5 sensors-25-04516-f005:**
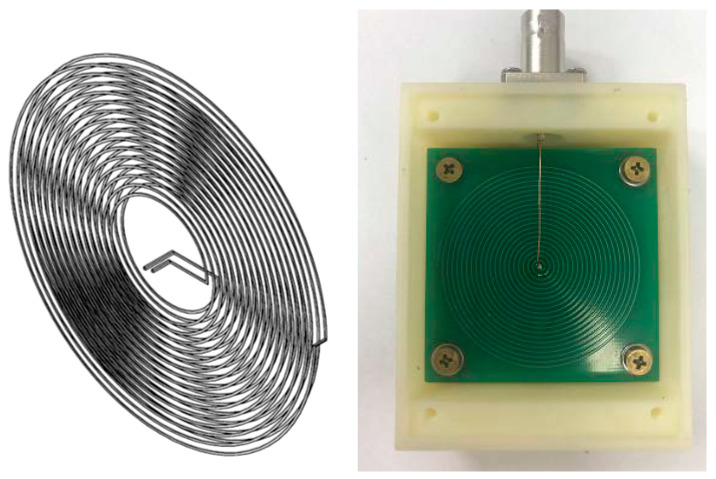
Structure and physical diagram of magnetic field sensor.

**Figure 6 sensors-25-04516-f006:**
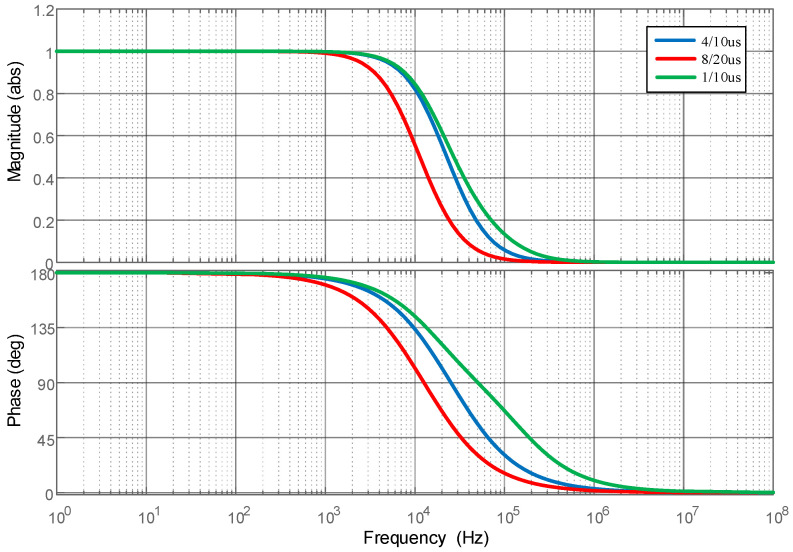
Frequency characteristic analysis of the Archimedean coil.

**Figure 7 sensors-25-04516-f007:**
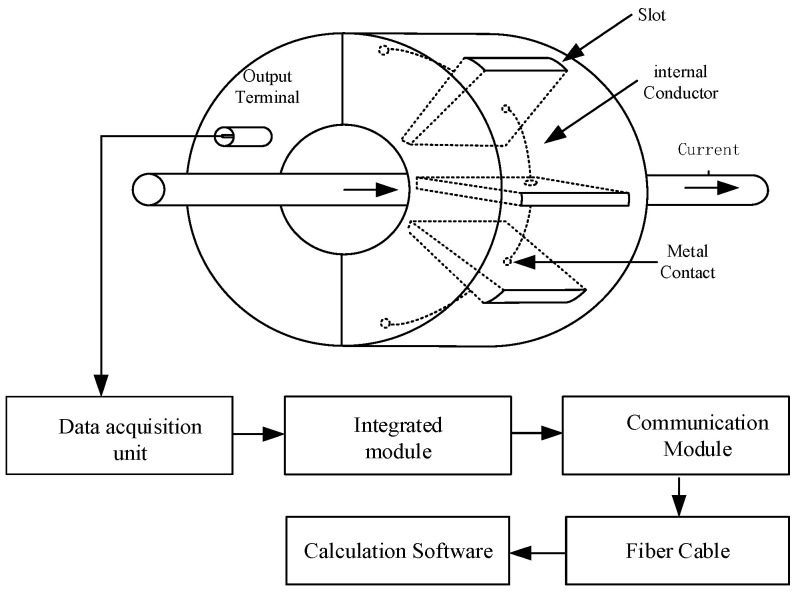
Schematic diagram of the magnetic field sensor with an array structure.

**Figure 8 sensors-25-04516-f008:**
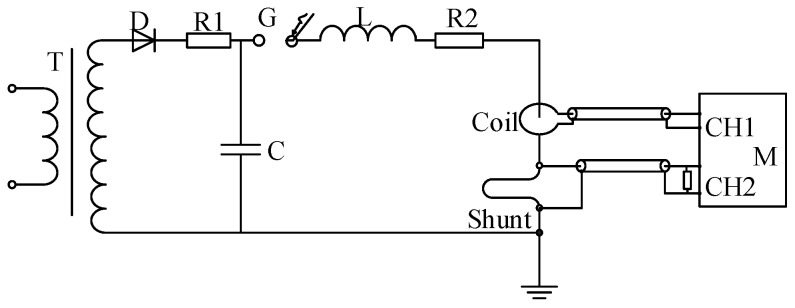
Schematic diagram for calibratoin test based on Rogowski coil. T—Transformer, D—High Voltage Silicon Stack, C—Charging Capacitor, G—Discharge Gap, L—Circuit Resistance, R2—Circuit Resistance, Coil—Rogowski Coil, Shunt—Current Divider, M—Digital Recorder.

**Figure 9 sensors-25-04516-f009:**
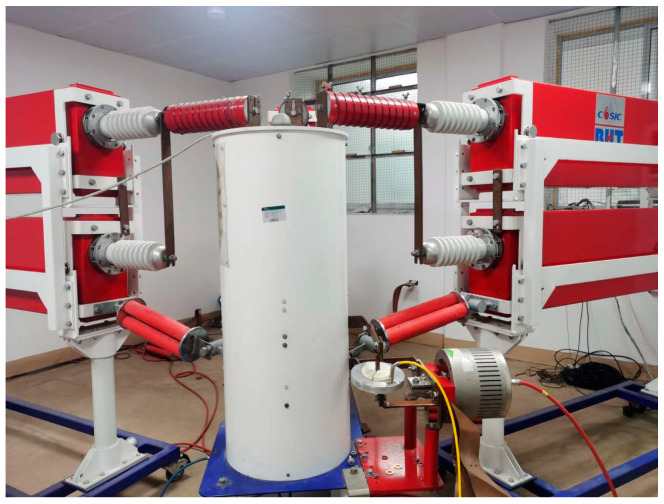
Experimental layout diagram for the Rigid Rogowski Coil.

**Figure 10 sensors-25-04516-f010:**
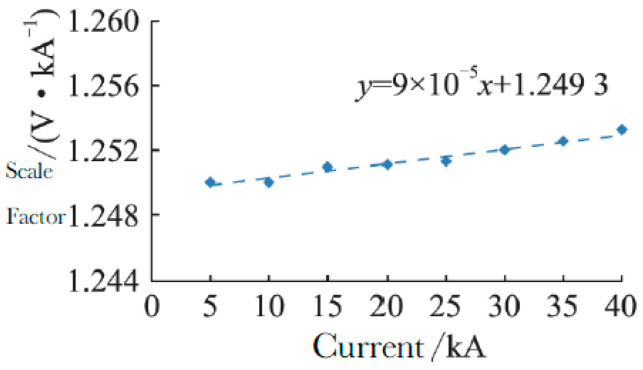
Scale factor calibration and extrapolation data.

**Figure 11 sensors-25-04516-f011:**
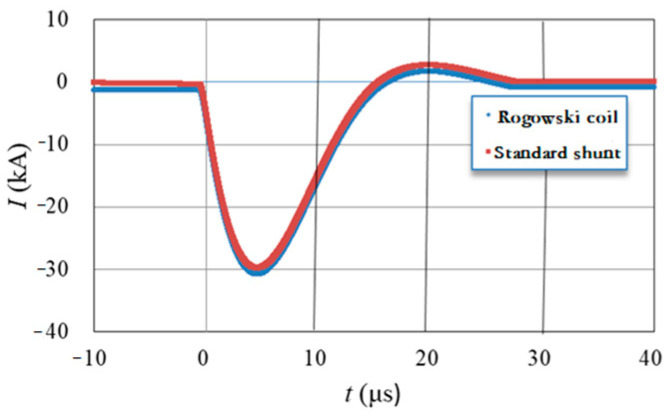
Calibration waveforms.

**Figure 12 sensors-25-04516-f012:**
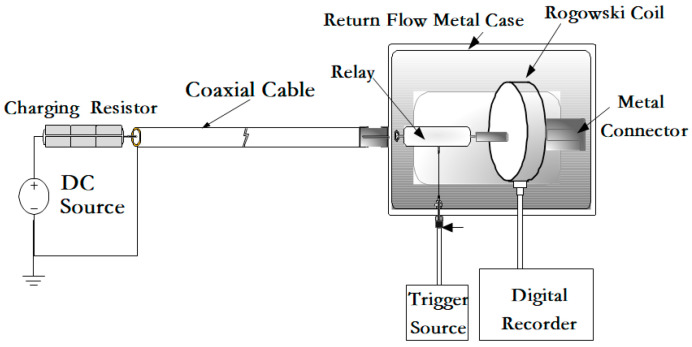
Schematic diagram of the Rogowski coil square wave response test circuit.

**Figure 13 sensors-25-04516-f013:**
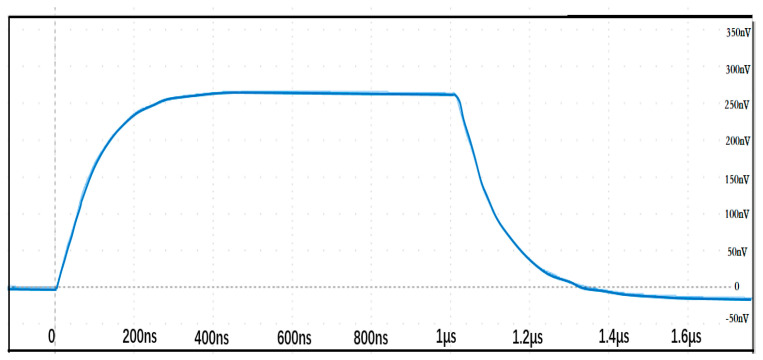
Step response waveform of the Rogowski coil.

**Figure 14 sensors-25-04516-f014:**
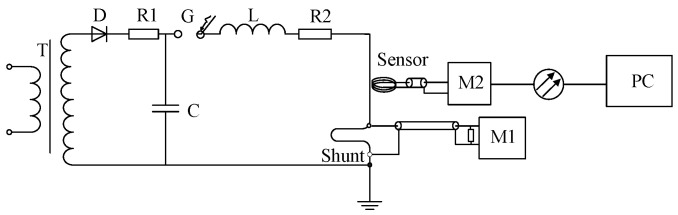
Schematic diagram for linearity test based on magnetic field sensor. T—Transformer, D—Silicon Stack, C—Charging Capacitor, G—Discharge Spark Gap L—Circuit Inductane, R2—Circuit Resistance, Sensor—Magnetic Field Sensor, Shunt—Shunt Resistor, M—Data Logger.

**Figure 15 sensors-25-04516-f015:**
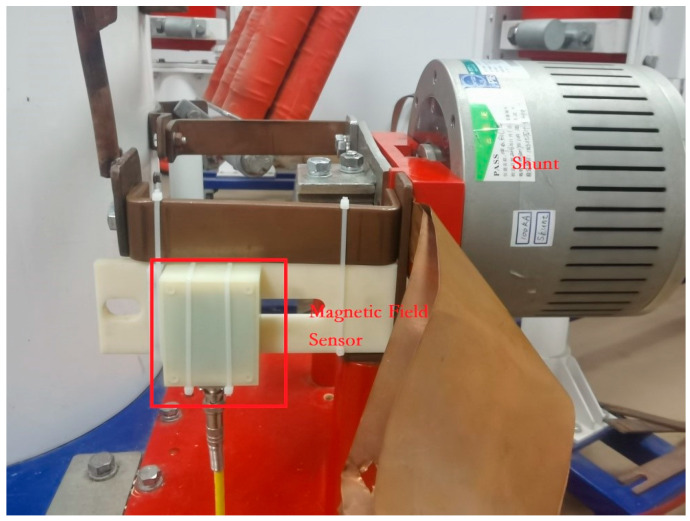
Experimental layout diagram.

**Figure 16 sensors-25-04516-f016:**
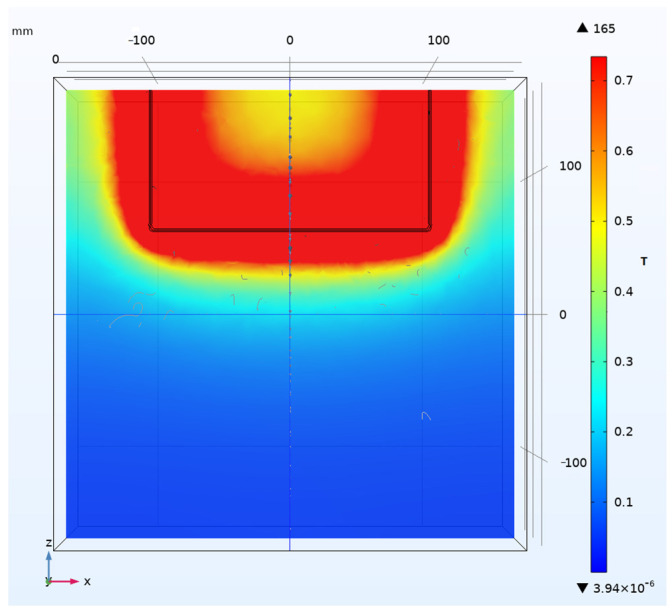
Simulation distribution map of magnetic field around current conductor.

**Figure 17 sensors-25-04516-f017:**
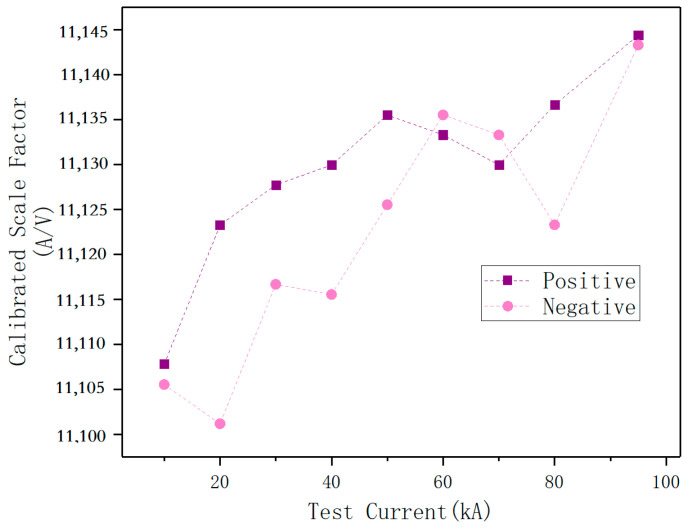
Scale Factor calibration results of magnetic field sensor.

**Table 1 sensors-25-04516-t001:** Circuit parameters of the designed Rogowski coil.

Coil Parameters	Value	Coil Parameters	Value
Self-inductance	3.04 μH	Distributed capacitance	2.32 (nF)
Low-frequency cutoff Frequency	80 Hz	High-frequency cutoff Frequency	1.8 (MHz)
Sensitivity	1.253 (V/kA)	Rise time	120 (ns)

**Table 2 sensors-25-04516-t002:** Calibration scale factor of the Rogowski coil within the 100 kA range.

Standard Current (kA)	Measured Current (kA)	Calibrated Scale Factor *k*_1_ (V/A)	Scale Factor of Fitting Curve *k*_2_ (V/A)	Relative Deviation (%)
49,624.1	49,632.0	1.2532	1.2529	−0.02
59,809.3	59,852.3	1.2539	1.2538	0.02
69,742.3	69,825.8	1.2545	1.2547	0.02
79,856.3	80,034.7	1.2558	1.2556	−0.02
89,769.2	89,998.5	1.2562	1.2565	−0.02
95,280.3	95,576.9	1.2569	1.2574	−0.02
49,624.1	49,818.2	1.2579	1.2583	−0.04

**Table 3 sensors-25-04516-t003:** Measurement uncertainty evaluation of the Rogowski coil.

Measurement Uncertainty Components	Parameter	Measurement Uncertainty Components	Parameter
Introduced by measurement repeatability *u*_A_	0.0002	Introduced by the short-term stability *u*_B3_	0.00063
Introduced by the standard measuring device *u*_B1_	0.002	Introduced by long-term stability *u*_B4_	0.001
Introduced by the linearity of the Rogowski coil’s scale factor *u*_B2_	0.001	/	/
Combined standard measurement uncertainty $u_{C}=\sqrt{u_{A}^{2}+u_{B1}^{2}+u_{B2}^{2}+u_{B3}^{2}+u_{B4}^{2}}$	2.53 × 10^−3^	Expanded measurement uncertainty (*k* = 2)	*U*_rel_ = 0.52%

## Data Availability

Dataset available on request from the authors.

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
