# Peer review of "Current Sensor with Optimized Linearity for Lightning Impulse Current Measurement"

_sensors, 2025, doi:10.3390/s25144516_

Round 1

Reviewer 1 Report

Comments and Suggestions for Authors
  1. 35 “According to IEC standards [8]” – ref.8 is not IEC standard. In general, when referring to a standard its number must be specified.
  2. There is no notation explanation of the notation in Eqs.1,2,3
  3. Abbreviations are not spelled.
  4. 2 caption is wrong.
  5. 100 “The 40kA lightning impulse current standard developed by CEPRI was used as calibration standard.” - pulse parameters must be specified and discussed.
  6. Explain how the measurement up to 100 kA were performed and can be trusted, while no standard is available.
  7. Fig 4 show data up to 40 kA, while measurements are made up to 100 kA. The missing data for 40 -100 kA interval are required.
  8. All characteristics of the Rogowski coil (bandwidth, nonlinearity, set time, etc), must be specified.
  9. L143: By designing a ring-shaped insulating support framework and arranging multiple sensor arrays, the effects of spatial co-directional electromagnetic interference can be effectively mitigated – explain how it is done practically.
  10. Provide characteristics of the magnetic sensors, such as band-width, sensitivity.
  11. 7 shows a double layer structure, which does not correspond to Figs. 5, 6. The model must be adjusted to the real device.
  12. 5, 9, 10, 12, etc.0 are not referred in the text niter properly explained.
  13. I_m, V in eq.4 are not explained.
  14. 4 neglects the heat transfer. It must be justified why is it possible.
  15. The values of the parameters in Eqs.4, 5 must be given.
  16. The actual dependence of the resistance on the temperature in Eq.5 must be provided.
  17. Operation of the circuit in Fig.13 must be explained and the parameters of the elements are given. Simulation of measurements is very desirable.
  18. 15 must be extended to include raw data of the Rogowski and shunt measurements, and the current measurement data. Why it is based on Rogowski and not thecalibrated currents data?
  19. According Fig. 8 the magnetic field sensors are placed around a conductor, while Fig.15 shows the sensor placed at a side. Explain how data are processed in this case.
  20. 18 What was sed as the reference? Add raw data for the sensor.
  21. Add error bars in Figs.15, 18
  22. L262 “the instantaneous temperature rise of the resistance material is calculated to be 52.4 ℃” – the measurements show much smaller increase. The difference must be explained.
  23. Data on the current calibration using the shunt temperature increase technique are missing.
  24. Conclusion (1) is not useful. It state that the model does not correspond to the reality. The wrong model must be adjusted or removed.
  25. Conclusion (2) states that the calibration of Rogowski is not complete. This puts into question the credibility of the whole work.
  26. Conclusion (3) is not justified. Detailed calibration of the magnetic sensor is required.

Author Response

Dear Reviewer,

Thank you for your careful review of the manuscript and for providing many valuable and detailed suggestions. Based on your feedback, I have revised the article accordingly. To improve the logical clarity of the paper, the content related to evaluating the linearity of the shunt through temperature rise calculations has been removed. The focus now lies on introducing the Rogowski coil and magnetic field sensor for measuring the linearity of impulse currents.

In the calibration experiments for the scale factors of the Rogowski coil and magnetic field sensor, the shunt is still employed as the reference standard. For the Rogowski coil, the effectiveness of the curve-fitting method has been validated in the text. For the magnetic field sensor, the method for determining its scale factor when the measured current exceeds 100 kA has been elaborated.

 The response to Reviewer's Comments is as follows:

  1. 35 “According to IEC standards [8]” – ref.8 is not IEC standard. In general, when referring to a standard its number must be specified.

Response1: The standard number IEC 62475 has been indicated in the text.

  1. There is no notation explanation of the notation in Eqs.1,2,3

Response2: The parameters in the formula have been annotated;

  1. Abbreviations are not spelled.

Response3: The full name of PCB has been provided.

  1. 2 caption is wrong.

Response4:  The caption of Figure 2 has been modified

  1. 100 “The 40kA lightning impulse current standard developed by CEPRI was used as calibration standard.” - pulse parameters must be specified and discussed.

Response5: This lightning current standard is applicable for measuring (4/10) μs and (8/20) μs lightning current waveforms, and supplementary explanations have been provided in the text.

  1. Explain how the measurement up to 100 kA were performed and can be trusted, while no standard is available.

Response6: In response to the reviewer’s feedback, the author has revised the manuscript. For the Rogowski coil, a 40 kA reference standard and a 100 kA reference standard were selected to calibrate the scale factors within the 40 kA and 100 kA ranges, respectively. By extrapolating the calibration curve based on the scale factors of the sensor within the 40 kA range, the extrapolated scale factors were compared with the experimentally calibrated values. The results confirm that the Rogowski coil demonstrates good linearity, as the extrapolated scale factors derived from the curve-fitting method show minimal deviation from the actual calibrated values.

  1. Fig 4 show data up to 40 kA, while measurements are made up to 100 kA. The missing data for 40 -100 kA interval are required.

Response7: The calibration factors for each current within the 40kA to 100kA range, have been supplemented in Figure

  1.  All characteristics of the Rogowski coil (bandwidth, nonlinearity, set time, etc), must be specified.   

Response8:  The aforementioned characteristic parameters of the developed Rogowski coil have been supplemented in the text.

  1. L143: By designing a ring-shaped insulating support framework and arranging multiple sensor arrays, the effects of spatial co-directional electromagnetic interference can be effectively mitigated – explain how it is done practically.

Response9:  The aforementioned characteristic parameters of the developed Rogowski coil have been supplemented in the text.

  1. Provide characteristics of the magnetic sensors, such as band-width, sensitivity.

Response10: The bandwidth parameters (3 Hz to 20 MHz) and the sensitivity of the magnetic field sensor, have been supplemented in the text.

  1. 7 shows a double layer structure, which does not correspond to Figs. 5, 6. The model must be adjusted to the real device.

Response11:  The sensor features a practical dual-layer structure. Figure 7 has been relocated before Figure 6 and the parametric analysis formulas. In the formulas, the right-hand side of the expression for the electrodynamic force E has been multiplied by 2.

  1. 5, 9, 10, 12, etc.0 are not referred in the text niter properly explained.

Response12:  The figures have been directly referenced in the main text.

  1. I_m, V in eq.4 are not explained.

Response13: The aforementioned parameters in the formula have been explained.

  1. 4 neglects the heat transfer. It must be justified why is it possible.

Response14:  Since the test current follows a (4/10) μs lightning current waveform, and considering the extremely short pulse width, the effect of heat dissipation on temperature rise is negligible.

  1. The values of the parameters in Eqs.4, 5 must be given.

Response15:  Revisions have been made.

  1. The actual dependence of the resistance on the temperature in Eq.5 must be provided.

Response16:  The temperature-dependent coefficient of resistance has been supplemented.

  1. Operation of the circuit in Fig.13 must be explained and the parameters of the elements are given. Simulation of measurements is very desirable.

Response17: The circuit component parameters of the (4/10) μs impulse current generator are provided in the text, along with the simulated waveform parameters.

  1. 15 must be extended to include raw data of the Rogowski and shunt measurements, and the current measurement data. Why it is based on Rogowski and not the calibrated currents data?

Response18: Data corresponding to the same current for both the Rogowski coil and the shunt have been added in Figure 15, where the shunt serves as the reference standard.

  1. According Fig. 8 the magnetic field sensors are placed around a conductor, while Fig.15 shows the sensor placed at a side. Explain how data are processed in this case.

Response19:  Figure 8 illustrates one configuration of the sensor array. However, due to the relatively uniform calibration test environment, an array-based layout was not adopted.

  1. 18 What was sed as the reference? Add raw data for the sensor.

Response20:  The content related to temperature rise measurements of the shunt has now been removed from the text.

  1. Add error bars in Figs.15, 18

Response21:  Revisions have been made.

  1. L262 “the instantaneous temperature rise of the resistance material is calculated to be 52.4 ℃” – the measurements show much smaller increase. The difference must be explained.

Response22: The content related to temperature rise measurements of the shunt has now been removed from the text.

  1. Data on the current calibration using the shunt temperature increase technique are missing.

Response23: The content related to temperature rise measurements of the shunt has now been removed from the text.

  1. Conclusion (1) is not useful. It state that the model does not correspond to the reality. The wrong model must be adjusted or removed.

Response24:  The content of Conclusion (1) has been revised.

  1. Conclusion (2) states that the calibration of Rogowski is not complete. This puts into question the credibility of the whole work.

Response25:   The content of Conclusion (2) has been revised.

  1. Conclusion (3) is not justified. Detailed calibration of the magnetic sensor is required.

Response26:  The content of Conclusion (3) has been revised.

We sincerely appreciate the reviewer’s constructive comments and would be grateful for any additional feedback to further improve this work.

Best Regards:

Wenting Li

Reviewer 2 Report

Comments and Suggestions for Authors

This article presents the research on scale factor calibration of impulse currents up to 100 kA. The linearity calibration results by three methods were compared, and the experimental results are obtained. The temperature rise calculation of the shunt depends on the resistance material parameters, dimensions, and waveform parameters of the test current. The developed rigid Rogowski coil and magnetic field sensor can used to calibrate the linearity parameters of the shunt. The linearity parameters are have good results.

There are two small questions:

  1. In Fig.4, why the Scale Factor is changed when current increase?
  2. Please give some explain or Instruction of Fig.10
  3. In Fig.15 and Fig.18, why there are some difference when current in position and negative? Are there come from the sensor design?
Comments on the Quality of English Language

no comments

Author Response

Dear Reviewer,

Thank you for your careful review of the manuscript and for providing many valuable and detailed suggestions. Based on your feedback, I have revised the article accordingly. To improve the logical clarity of the paper, the content related to evaluating the linearity of the shunt through temperature rise calculations has been removed. The focus now lies on introducing the Rogowski coil and magnetic field sensor for measuring the linearity of impulse currents.

In the calibration experiments for the scale factors of the Rogowski coil and magnetic field sensor, the shunt is still employed as the reference standard. For the Rogowski coil, the effectiveness of the curve-fitting method has been validated in the text. For the magnetic field sensor, the method for determining its scale factor when the measured current exceeds 100 kA has been elaborated.

 The response to Reviewer's Comments is as follows:

Comments1: In Fig.4, why the Scale Factor is changed when current increase?

Response 1: The experimental data in Figure 4 are based on calibration test results. In the Rogowski coil measurement system, besides the PCB measurement coil, there exists a backend integration circuit. The parameters of resistive and capacitive components in the measurement system exhibit slight variations under different current levels, leading to corresponding changes in the scale factor. The trend shown in Figure 4 demonstrates a gradual increase in the scale factor with current, though the magnitude of variation remains minimal.

Comments2: Please give some explain or Instruction of Fig.10

In the revised manuscript, Figure 10 presents the scale factor of the sensor calibrated using a standard shunt within the 100 kA test current range. The paper elaborates on how to determine the scale factor of the magnetic field sensor when the test current is further increased, and demonstrates its application in calibrating the linearity of other high-current sensors using this established scale factor.

Comments3: In Fig.15 and Fig.18, why there are some difference when current in position and negative? Are there come from the sensor design?

Response3: Figures 15 and 18 present the calibrated scale factors of two impulse current sensors under positive and negative polarities. The scale factors exhibit slight discrepancies between polarities. These observed differences may originate from the integration circuit in the sensor's backend. Specifically, the DC offset voltage of the operational amplifier in the integration circuit introduces asymmetric integration errors under positive and negative input signals, leading to proportional deviations in the voltage amplitudes between positive and negative current signals after integration.

Reviewer 3 Report

Comments and Suggestions for Authors

The paper addresses an interesting technical topic, but in its current form does not meet the standards of a scientific paper. The authors should describe more clearly the novelty of their research. A detailed methodology of the calibration process, uncertainty analysis (statistical analysis) and references to existing literature are missing. In particular, authors should note the following comments:

1) the abstract should start differently, the first sentence sounds like it's taken out of context,

2) the introduction lacks a literature review and the author's solution is placed against the background of the state of knowledge, it is difficult to indicate whether the work contributes anything new,

3) the work is not entirely scientific in nature, it is more like a report on a completed exercise.

Author Response

Comments1:  the abstract should start differently, the first sentence sounds like it's taken out of context,  

Response1: The abstract has been revised;

Comments2:  the introduction lacks a literature review and the author's solution is placed against the background of the state of knowledge, it is difficult to indicate whether the work contributes anything new,

Response 2: A literature review has been added to the introduction section;

Comments3:  the work is not entirely scientific in nature, it is more like a report on a completed exercise.

Response3: The article’s section structure and content have been restructured and revised. Please review the manuscript again.

Reviewer 4 Report

Comments and Suggestions for Authors
  1. Please give the full names for all abbreviations upon their first appearance in the manuscript, such as IEC, PTB, CEPRI, ICMS….

  2. Please define all symbols used in the equations.

  3. Please add a legend to Fig.4. 

  4. Why not consider using a Rogowski coil for directly measuring current up to 100 kA, rather than relying on extrapolated data?

  5. Is it possible to use other types of magnetic field sensor, such as hall sensors?

  6. Please use English legend for Fig. 9.

  7. Some figures (e.g., Figures 8, 9, and 10) are not clearly described or referenced in the main text. Please ensure that all figures are properly introduced and discussed.

Author Response

Dear Reviewer,

Thank you for your careful review of the manuscript and for providing many valuable and detailed suggestions. Based on your feedback, I have revised the article accordingly. To improve the logical clarity of the paper, the content related to evaluating the linearity of the shunt through temperature rise calculations has been removed. The focus now lies on introducing the Rogowski coil and magnetic field sensor for measuring the linearity of impulse currents.

In the calibration experiments for the scale factors of the Rogowski coil and magnetic field sensor, the shunt is still employed as the reference standard. For the Rogowski coil, the effectiveness of the curve-fitting method has been validated in the text. For the magnetic field sensor, the method for determining its scale factor when the measured current exceeds 100 kA has been elaborated.

 The response to Reviewer's Comments is as follows:

Comments1: Please give the full names for all abbreviations upon their first appearance in the manuscript, such as IEC, PTB, CEPRI, ICMS….

Response1: All acronyms in the text have been annotated with their full names upon their first occurrence.

Comments2: Please define all symbols used in the equations.

Response2: Definitions for all symbols used in equations have been supplemented.

Comments3:Please add a legend to Fig.4. 

Response3: Figure 4 has been supplemented with detailed experimental conditions.

Comments4:Why not consider using a Rogowski coil for directly measuring current up to 100 kA, rather than relying on extrapolated data?

Response4: To validate whether the curve-fitting method is applicable for calibrating scale factors at higher current levels, the revised manuscript now compares curve-fitting results with calibrated values within the 40–100 kA range. This verification demonstrates the feasibility of extrapolating the calibration method beyond 100 kA, serving as a reference for ultra-high-current measurements.

Comments5:Is it possible to use other types of magnetic field sensor, such as hall sensors?

Response5: Alternative sensor types (e.g., Hall sensors) have not been selected in the current study due to bandwidth limitations (up to 10 MHz) and their insufficient dynamic response for steep-front impulse currents (e.g., 1.2/50 μs waveforms).

Comments6:Please use English legend for Fig. 9.

Response6: All figure captions (including Figure 9) have been updated to English.

Comments7:Some figures (e.g., Figures 8, 9, and 10) are not clearly described or referenced in the main text. Please ensure that all figures are properly introduced and discussed.

Response7: In-text citations for Figures 8–10 have been added to ensure proper contextualization.

Round 2

Reviewer 1 Report

Comments and Suggestions for Authors

The corrections are fine, the manuscript is suitable for publication.

Author Response

Dear Reviewer,

Thank you for your valuable and detailed suggestions on this paper and for your affirmation of my revised version. Based on the editor's suggestions, I have optimized the introduction and conclusion sections and supplemented relevant references. Please review the manuscript again.

Reviewer 3 Report

Comments and Suggestions for Authors The authors have responded to all comments in an exhaustive manner. In my opinion, the article can be published in its current form.

Author Response

(The authors gave the same response as above.)
